# Barley sodium content is regulated by natural variants of the Na⁺ transporter *HvHKT1;5*

Kelly Houston[1], Jiaen Qiu [2,3], Stefanie Wege[2,3], Maria Hrmova [3,4], Helena Oakey [3], Yue Qu [2,3], Pauline Smith[1], Apriadi Situmorang[2,3], Malcolm Macaulay[1], Paulina Flis [5], Micha Bayer[1], Stuart Roy [3,6], Claire Halpin [7], Joanne Russell[1], Miriam Schreiber[1], Caitlin Byrt [2,3,8], Matt Gilliham [2,3✉], David E. Salt [5✉] & Robbie Waugh [1,3,7✉]

During plant growth, sodium (Na⁺) in the soil is transported via the xylem from the root to the shoot. While excess Na⁺ is toxic to most plants, non-toxic concentrations have been shown to improve crop yields under certain conditions, such as when soil K⁺ is low. We quantified grain Na⁺ across a barley genome-wide association study panel grown under non-saline conditions and identified variants of a Class 1 *HIGH-AFFINITY-POTASSIUM-TRANSPORTER* (*HvHKT1;5*)-encoding gene responsible for Na⁺ content variation under these conditions. A leucine to proline substitution at position 189 (L189P) in HvHKT1;5 disturbs its characteristic plasma membrane localisation and disrupts Na⁺ transport. Under low and moderate soil Na⁺, genotypes containing HvHKT1:5$_{P189}$ accumulate high concentrations of Na⁺ but exhibit no evidence of toxicity. As the frequency of HvHKT1:5$_{P189}$ increases significantly in cultivated European germplasm, we cautiously speculate that this non-functional variant may enhance yield potential in non-saline environments, possibly by offsetting limitations of low available K⁺.

[1] Cell and Molecular Sciences, The James Hutton Institute, Errol Road Invergowrie, Dundee DD2 5DA Scotland, UK. [2] ARC Centre of Excellence in Plant Energy Biology, University of Adelaide, Waite Campus, Glen Osmond, SA 5064, Australia. [3] School of Agriculture and Wine & Waite Research Institute, University of Adelaide, Waite Campus, Glen Osmond, SA 5064, Australia. [4] School of Life Science, Huaiyin Normal University, 223300 Huaian, China. [5] Future Food Beacon of Excellence and the School of Biosciences, University of Nottingham, Nottingham NG7 2RD, UK. [6] ARC Industrial Transformation Research Hub for Wheat in a Hot Dry Climate, University of Adelaide, Waite Campus, Glen Osmond, SA 5064, Australia. [7] School of Life Sciences, University of Dundee, Dow Street, Dundee DD1 5EH Scotland, UK. [8] Research School of Biology, 46 Sullivans Creek Road, The Australian National University, Canberra, ACT 2601, Australia. ✉email: matthew.gilliham@adelaide.edu.au; David.Salt@nottingham.ac.uk; robbie.waugh@hutton.ac.uk

In C3 plants, such as *Arabidopsis*, wheat, rice, and barley, sodium ($Na^+$) is non-essential for plant growth and development[1]. While halophytes thrive on high $Na^+$ containing soils[2], for glycophytes, including our major cereal crops, $Na^+$ becomes toxic when present above certain species-specific threshold concentrations. Intriguingly, many crops, including barley, have been shown to benefit from intermediate (non-toxic) concentrations of $Na^+$, a situation that is particularly evident when levels of $K^+$ in the soil are low or unavailable[1,3–10]. Extensive historical evidence supports a requirement for nontoxic concentrations of $Na^+$ to achieve maximal biomass growth in a wide range of plants[11]. As such it has been suggested that $Na^+$ is a functional nutrient, proposed to be capable of substituting for many of the essential roles that $K^+$ ions play in plant nutrition including osmoregulation and enzyme activation[1].

Despite the reported positive attributes of $Na^+$, by far the majority of studies in the more recent literature focus on the negative impacts of $Na^+$ (i.e. salinity) on plant growth[12,13]. These latter investigations generally seek to explore the possible mechanisms that explain how tolerance to excess $Na^+$ can be achieved. They commonly revolve around $Na^+$ exclusion from the transpiration stream via active removal in the root, the partitioning of excess $Na^+$ into root and shoot vacuoles to protect the $Na^+$ sensitive photosynthetic tissues of the shoot, or the energy balance associated with active tolerance mechanisms[14,15]. While understanding how to enable crops to grow more efficiently in the expanding saline environments across the globe is highly relevant, it remains important to note that the majority of temperate cereal crop production is achieved on nonsaline soils. Given the demonstrated benefits of $Na^+$ as a functional nutrient, we have taken a combined genetic and functional approach to explore the extent and causes of natural variation in $Na^+$ content in barley grown in nonsaline soils. Our data lead us to speculate, cautiously, that high $Na^+$ accumulation may be a positive trait in the nonsaline conditions typical of high production agricultural environments.

## Results

**Barley grain $Na^+$ content is under genetic control.** We used Inductively Coupled Plasma Mass Spectrometry (ICP-MS) to quantify sodium ($Na^+$) and potassium ($K^+$) content of wholegrain samples from five biological and five technical replicates of a small barley genome-wide association study (GWAS) panel comprised of 131 elite two-row spring genotypes. Elite barley germplasm displays significant population structure due to both winter/spring growth habit and two- versus six-row inflorescence architecture[16]. By focusing only on the two-row spring barley genepool we effectively remove confounding effects of underlying population structure[17]. All plants were grown under optimal, nonsaline conditions. We observed an approximate six-fold variation in grain $Na^+$ (16.07 to 98.76 ppm) and greater than twofold variation in $K^+$ (2459–5562 ppm) contents (Fig. 1a and Supplementary Data 1). We then used barley 50k iSelect SNP genotypic data collected from all 131 genotypes to conduct GWAS[18]. For grain $Na^+$ content we observed a single highly significant association on the bottom of chromosome 4HL for the marker SCRI_RS_142792, with a $-\log10(p) = 11.8$, FDR adjusted $p$-value $= 2.2E-08$, and an $R^2 = 0.45$ (Fig. 1b) consistent with prior genetic analyses of shoot $Na^+$ content[19–21]. The moderately high $R^2$ value implies other loci and mechanisms are also involved in this trait. No significant genetic association was observed for grain $K^+$ content at this locus (Fig. 1c). The significantly associated region spanned ~6.6 Mb, from 638,211,825 nt to 644,818,273 nt on the barley 4 H physical map[22] and contained 247 gene models (Supplementary Data 2). Common SNPs

between the 50k and 9 K iSelect platforms[17,18] aligned this association with a region recently shown to contain *HvHKT1;5*[23], and in the barley genome sequence HORVU4Hr1G087960, 0.58 kb from the top scoring SNP (SCRI_RS_142792), was annotated as a homolog of *OsHKT1;5*. Considering this and previous functional studies[24–27], we conclude that HORVU4Hr1G087960 is the $Na^+$ specific transporter *HvHKT1;5*.

As *HvHKT1;5* is a clear candidate for causing the observed phenotypic variation we PCR-sequenced this gene from all 131 genotypes included in the GWAS. We observed 10 nonsynonymous SNPs that defined three haplotypes (defined as $Na^+_{HAP1}$, $Na^+_{HAP2}$, and $Na^+_{HAP3}$) (Fig. 1d–f). $Na^+_{HAP1}$ and $Na^+_{HAP2}$ correspond exactly to the HGB haplotype very recently described[21]. In our GWAS panel there was no association between *HvHKT1;5* haplotype and population structure. We observed a significant difference in mean grain $Na^+$ content between genotypes containing the low grain $Na^+$ haplotypes, $Na^+_{HAP1}$ ($M = 28.6$, SD $\pm 72$) and $Na^+_{HAP2}$ ($M = 22.2$, SD $\pm 6.48$), and the high grain $Na^+_{HAP3}$ allele ($M = 51.6$, SD $\pm 15.2$); t(52) $= 9.07$, $p = 2.66E-12$) (Fig. 1g). There was no significant difference between $Na^+_{HAP1}$ and $Na^+_{HAP2}$ ($M = 28.6$, SD $\pm 7.72$, $M = 22.2$, SD $\pm 6.48$); t(2) $= 1.66$, $p = 0.23$). Lines containing $Na^+_{HAP3}$ contain an average increase in grain $Na^+$ content of 1.8-fold over $Na^+_{HAP1}$ and $Na^+_{HAP2}$. Of the 10 nonsynonymous SNPs, six in complete linkage disequilibrium (LD) altered amino-acid residues (S56N, Q102E, N130K, L189P, I416V, and N438S) that differentiated the two low $Na^+$ haplotypes ($Na^+_{HAP1}$ and $Na^+_{HAP2}$) from the high $Na^+$ haplotype ($Na^+_{HAP3}$). The remaining four (P133Q, I223V, T377A, and H402Q) differentiated low $Na^+_{HAP1}$ and high $Na^+_{HAP3}$ from $Na^+_{HAP2}$ (Fig. 1d, e).

**Transcript abundance varies between *HvHKT1;5* haplotypes.** As variation in transcript abundance among *HvHKT1;5* haplotypes has previously been implicated in determining variation in shoot $Na^+$ content[21], we selected genotypes that were representative of $Na^+_{HAP1}$ (cv. Golden Promise), $Na^+_{HAP2}$ (cv. Viivi), and $Na^+_{HAP3}$ (cv. Morex) and quantified *HvHKT1;5* transcript abundance by qRT-PCR in a range of tissues after growth without added $Na^+$ (Fig. 2). Golden Promise and Morex were chosen because they had available genome assemblies and publicly available transcript data for a variety of tissues[22,28]. We observed significant differences ($P < 0.05$) in the normalised gene expression in each haplotype across tissues. In general terms, *HvHKT1;5* was more highly expressed in roots compared to shoots, and in low $Na^+_{HAP1}$ and $Na^+_{HAP2}$ compared to high $Na^+_{HAP3}$. The highest overall expression was observed in the maturation zone of the roots in cv. Viivi ($Na^+_{HAP2}$) with in situ hybridisations using sections from this region showing that *HvHKT1;5* was predominantly expressed in the xylem parenchyma and endodermal cells adjacent to the xylem vessels (Fig. 2c)[24,25,29]. These results agree with previous reports that *HvHKT1;5* transcript abundance in roots is inversely correlated with grain $Na^+$ content.

**A single substitution in HvHKT1;5$_{HAP3}$ disrupts $Na^+$ transport function.** Despite the current lack of evidence for natural functional variation in HvHKT1;5[21] we were interested in testing whether the observed haplotypes influenced in vivo $Na^+$ transport properties. We assembled and independently tested constructs expressing $Na^+_{HAP1}$, $Na^+_{HAP2}$, or $Na^+_{HAP3}$ in *Xenopus laevis* oocytes using two-electrode voltage-clamp (TEVC) experiments. Oocytes injected with cRNA of $Na^+_{HAP1}$ (or $Na^+_{HAP2}$, Supplementary Fig. 1) showed significant inward currents in the presence of external $Na^+$ but not $K^+$ (Fig. 3) consistent with *HvHKT1;5* being a $Na^+$-specific transport protein. With the external $Na^+$ concentration increased from 1 mM to

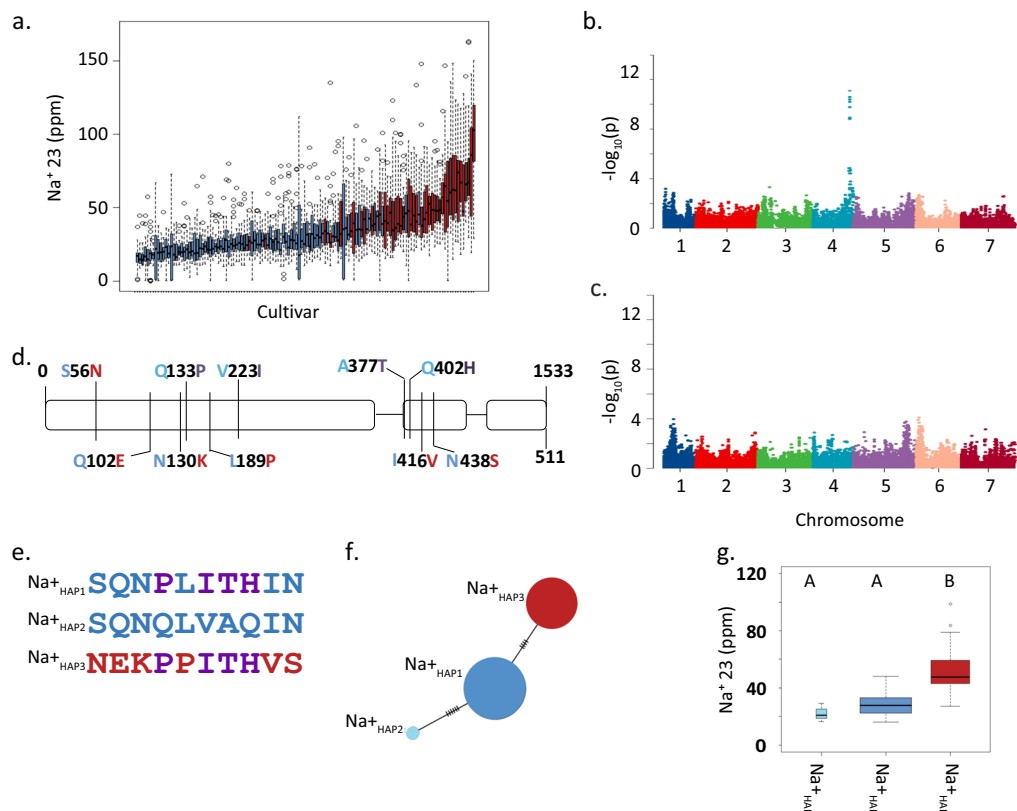

**Fig. 1 HvHKT1;5 haplotypes influence grain Na⁺ content. a** Box plots of of grain sodium content quantified using ICP-MS, $n = 5$ biologically independent samples. Blue bars represent lines containing L189 and red bars represent genotypes containing P189. **b** Manhattan plot of GWAS of grain Na⁺ 23 content, FDR threshold = −log 10(P)=6.02. **c** Manhattan plot of GWAS of grain K⁺ content using, FDR threshold = −log 10(P)=6.02. **d** Gene structure and nonsynonymous polymorphisms coloured according to haplotype shown in **e**. **e** Haplotype summary based on nonsynonymous SNPs. **f** Haplotype analysis of nonsynonymous SNPs. Circles scaled to number of individuals sharing the haplotype, and short lines represent number of SNPs differentiating haplotype groups. **g** Box plots of grain Na⁺ contents in the three haplotypes, $n = 3$, 73, and 41, width of plots scaled to number of individuals from each haplotype. Different letters above boxes indicate significant difference in grain sodium content $p < 0.05$, same letters indicate no significant difference using this threshold. For both sets of boxplots the horizontal bar of the boxplot shows the median, the box delineates the first and third quartile, and the whiskers show ±1.5 × IQR.

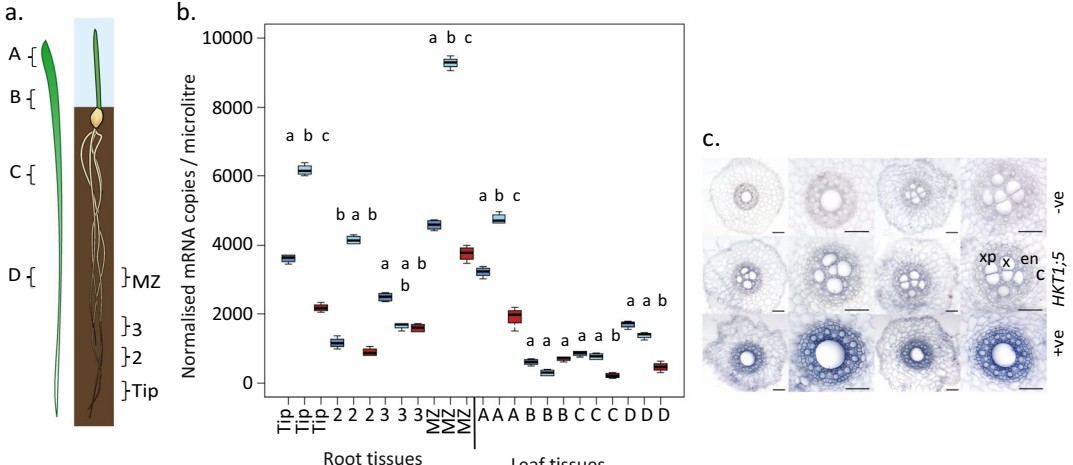

**Fig. 2 Spatial expression patterns of *HvHKT1;5* haplotypes. a** Sections used for quantification of transcript abundance (after Burton et al., 2004[47]). **b** Quantification of *HvHKT1;5* in root (Tip, 2, 3, and MZ) and shoot (A, B, C, and D) tissues from 12-day-old barley plants. Dark blue—Na⁺HAP1 (cv. Golden Promise), pale blue = Na⁺HAP2 (cv. Vivii), red = Na⁺HAP3 (cv. Morex) differences from after carrying out an ANOVA followed by Tukey HSD within tissue type, same letters in lower case indicate no significant difference using $P < 0.05$. For boxplots the horizontal bar of the boxplot shows the median, the box delineates the first and third quartile, and the whiskers show ±1.5 × IQR. **c** In situ localisation of *HvHKT1;5* in 2-week-old barley root tissue from cv. Golden Promise, Na⁺HAP1 (grown without added NaCl). Top: negative controls with no RT (reverse transcription), Middle *HvHKT1;5* with c cortex, en endodermis, x xylem, xp xylem parenchyma labelled in red. Bottom *Hv18S* rRNA (positive control). Scale bars, 100 μm.

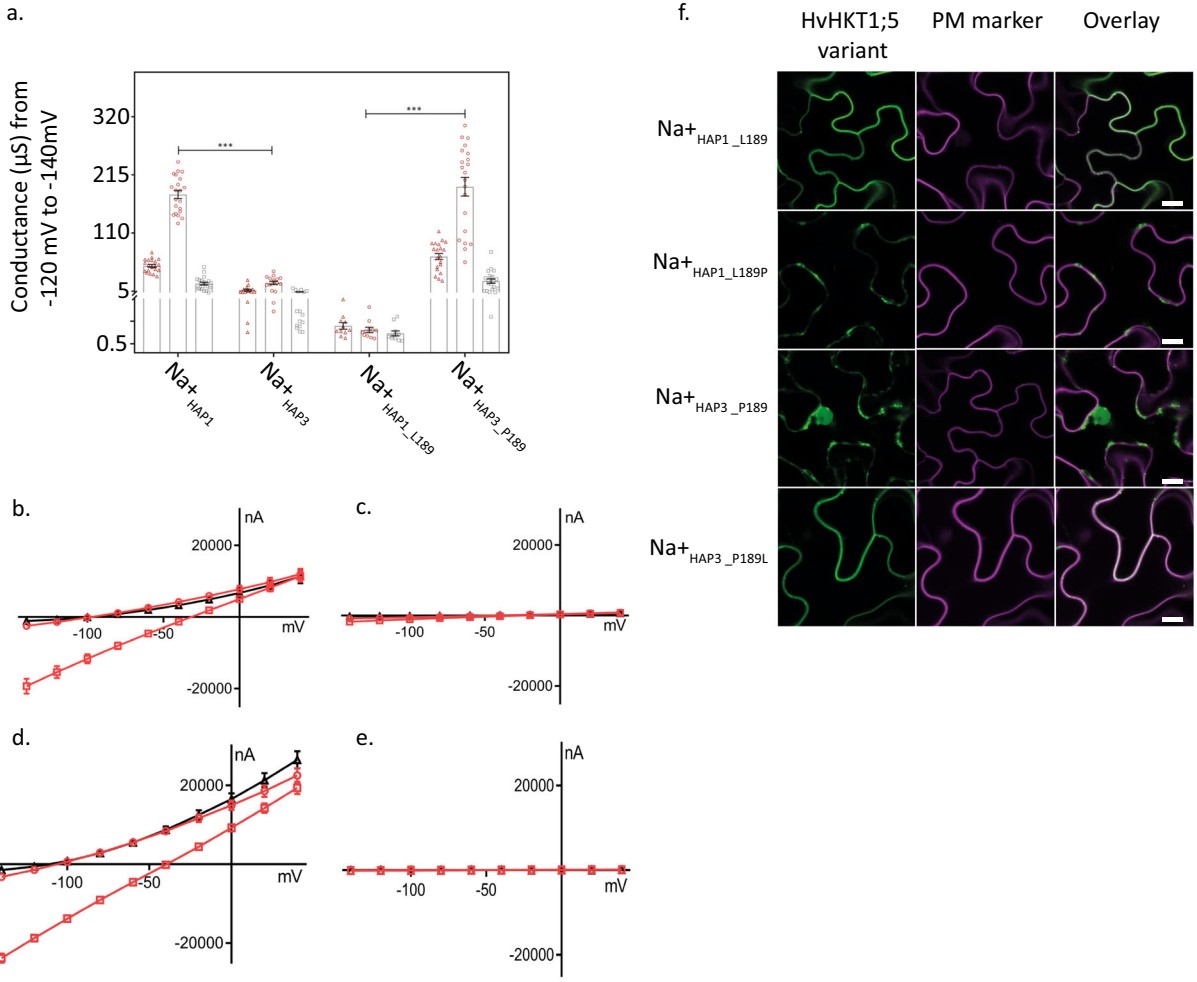

**Fig. 3 Heterologous expression of HvHKT1;5 variants in Xenopus laevis oocytes and Nicotiana benthamiana epidermal cells. a** Cation ($Na^+$ and $K^+$) conductance ($-140\,mV$ to $-120\,mV$) from HvHKT1;5 haplotypes cRNA-injected oocytes perfused with $1\,mM$ (open red triangles), $30\,mM\ Na^+$ (open red squares) and $30\,mM\ K^+$ (open grey squares). Data are means ± SEM of currents, $n = 11$–21, each triangle or square represents a single sample, (***$P <$ 0.001), combined from three independent experiments. **b–e** Representative I–V curves of cRNA from HvHKT1;5 haplotypes injected into *Xenopus laevis* oocytes ($n = 4$) clamped at $-140\,mV$ to $40\,mV$ in $Na^+$ or $K^+$ solutions· Red circles ($1\,mM\ Na^+$), red squares ($30\,mM\ Na^+$), black triangles ($30\,mM\ K^+$). **b** $Na^+_{HAP1\_L189}$; **c** $Na^+_{HAP3\_P189}$; **d** $Na^+_{HAP3\_L189}$; **e** $Na^+_{HAP3\_P189}$. **f** Transient co-expression of GFP-HvHKT1;5 variants with CBL1n-RFP plasma membrane marker in *Nicotiana benthamiana* leaf epidermal cells. GFP signal in the left panel (green), RFP-signal in the middle (magenta), overlay on the right (colocalisation of green and magenta signals appears in white). Scale bars = $10\,\mu m$.

$30\,mM$, a two-fold increase in $Na^+$ conductance was observed (Fig. 3). For $Na^+_{HAP3}$, $Na^+$ and $K^+$ elicited currents were similar to water-injected controls with the conductance unaltered when external $Na^+$ concentration was increased (Fig. 3), indicating that $Na^+_{HAP3}$ was severely compromised in its ability to transport $Na^+$ across the plasma membrane.

Publicly available data for HKT1;5 led us to focus on four single amino-acid residue changes as potentially causal for compromised transport (N57S, P189L, V416I, and S438N)[21,29]. We swapped these candidate amino acids residues individually into the compromised high $Na^+_{HAP3}$ and quantified their impact on $Na^+$ conductance by TEVC in the oocyte system. In comparison to high $Na^+_{HAP3}$, $Na^+_{HAP3\_L189}$ showed a $Na^+$-dependent conductance that was comparable to low $Na^+_{HAP1}$ (Fig. 3). The reciprocal substitution (P189) into low $Na^+_{HAP1}$ showed significantly reduced $Na^+$ conductance, comparable to high $Na^+_{HAP3}$ (Fig. 3). No other substitution converted a low $Na^+$ haplotype to a high $Na^+$ haplotype or vice versa; however, the $Na^+_{HAP1\_V416}$, reduced but did not abolish the $Na^+$-dependent conductance of $Na^+_{HAP1}$ (Supplementary Fig. 1).

These data support the conclusion that the naturally occurring P189 amino-acid residue in $Na^+_{HAP3}$ compromises the function of HvHKT1;5, and that certain variants (e.g. $Na^+_{HAP1\_V416}$) can also affect $Na^+$ dependent conductance in TEVC experiments.

**HvHKT1;5$_{HAP3}$ does not localise to the plasma membrane.** Consistent with their role in $Na^+$ retrieval from the xylem sap, HKT1;5 proteins have been previously shown to localise specifically to the plasma membrane (PM)[30]. We were therefore interested in whether the observed functional variation had consequences for HvHKT1;5 subcellular localisation. We transiently co-expressed N-terminally GFP-tagged HvHKT1;5$_{HAP3\_L189P}$ variants with a plasma membrane (PM)-marker in *Nicotiana benthamiana* epidermal cells. Confocal imaging revealed that the low $Na^+$ variant HvHKT1;5$_{HAP3\_L189}$ was almost exclusively localised at the PM (Fig. 3). However, the high $Na^+$ HvHKT1;5$_{HAP3\_P189}$ did not colocalise with the PM-marker; the GFP signal was instead localised to internal cell structures (Fig. 3). Introduction of P189 into HvHKT1;5$_{HAP1}$ phenocopied the GFP-signal pattern of cells transformed with HvHKT1;5$_{HAP3}$ (Fig. 3). This GFP-signal pattern

in cells expressing HvHKT1;5 haplotypes harbouring P189 may suggest protein degradation.

To explore this further we constructed 3D molecular models of HvHKT1;5$_{HAP3\_L189}$ and HvHKT1;5$_{HAP3\_P189}$ in complex with Na$^+$ using the *B. subtilis* KtrB K$^+$ transporter (Protein Data Bank genotype 4J7C, chain I) as a template with K$^+$ substituted by Na$^+$[31,32] (Supplementary Fig. 2). In the structural models, detailed analysis of the micro-environments around α-helices 4 and 5 revealed that the α-helix 4 of low Na$^+$ allele HvHKT1;5$_{HAP3\_L189}$ established a network of four polar contacts at separations between 2.7 Å to 3.1 Å with A185, V186, Y192, and S193 neighbouring residues. However, these were not formed in high Na$^+$ HvHKT1;5$_{HAP3\_P189}$, which only established two polar contacts at separations between 2.5 and 2.7 Å with S193. We observed a positive correlation between the structural characteristics of α-helices 4 and 5 (trends in angles based on α-helical planes), differences in Gibbs free energies of forward (P189L) and reverse (L189P) mutations, and the ability to produce Na$^+$ fluxes across oocyte membranes. Combined with our previous observations we hypothesise that P189 in HvHKT1;5 does affect protein structure, potentially triggering protein degradation prior to insertion into the plasma membrane thereby reducing Na$^+$ retrieval from the xylem with bulk flow ultimately elevating Na$^+$ in the grain.

**The impact of *HvHKT1;5* haplotypes under salt stress**. While our original data were collected from plants grown under optimal conditions, most recent reports in the literature focus on the impact of variation at HKT1;5 on natural tolerance to growth in saline environments[33–35]. We therefore explored the impact of *HvHKT1;5* variants on a range of phenotypic traits after growth in 0, 150, and 250 mM added NaCl (Fig. 4a). While we observed confounding between allele, haplotype, and line (see full analysis given in Supplementary dataset 1), we can nevertheless conclude that grain Na$^+$ content is influenced significantly by both allele (L189P) and treatment (NaCl); lines containing the functionally compromised P189 allele accumulate higher concentrations of Na$^+$ than lines containing L189 and show a larger difference between control and salt treatments. All lines with L189 accumulated less Na$^+$ in the grain than those with P189, with one genotype, Maris Mink, having especially high grain Na$^+$. This line had the second highest concentration of grain Na$^+$ when grown as part of the GWAS panel (Supplementary Data 1). Despite the higher Na$^+$ content, no clear detrimental effect of P189 was observed on vegetative biomass (Fig. 4b). NaCl at 250 mM had a strong and consistent negative influence on total biomass across all lines regardless of *HVHKT1;5* haplotype. There was no influence of L189P on grain K$^+$ content ($P > 0.05$), although there was an effect of treatment on this trait ($P = 0.02$) (Supplementary Fig. 3). As previous studies of HKT1;5 s generally focus on shoot Na$^+$, we also examined shoots from the same plants. Again, we observed that an interaction between allele and treatment significantly influenced leaf Na$^+$ content ($P < 0.001$), with lines containing L189 accumulating less Na$^+$ in leaf tissue than those with P189 (Supplementary Fig. 4, Supplementary Data 5), mirroring our observations in grain.

**HvHKT1;5$_{HAP3}$ frequency increases significantly in elite barley**. Previously observed associations between *AtHKT1* alleles and the environment[36] prompted us to explore whether barley *HvHKT1;5* haplotypes had any obvious evolutionary or ecological significance. We identified and downloaded orthologs of *HKT1;5* and aligned the retrieved protein sequences with MUSCLE[37]. Based on the available sequences only barley genotypes contained the L189P substitution in HKT1;5 despite comparing amino-acid

sequences of 11 different species (Supplementary Fig. 5). To explore the origin and distribution of the P189 variant, we then PCR-sequenced *HvHKT1;5* from a collection of 73 georeferenced wild barley (*H. spontaneum*) genotypes from the fertile crescent[38]. This revealed 19 additional nonsynonymous SNPs defining 27 haplotypes (Supplementary Data 3) that are distinct from those observed in the GWAS panel, with the exception of Na$^+$$_{HAP3}$ that was found in one genotype, FT064, originating from southern Israel (latitude = 31.35, longitude = 35.12) (Supplementary Figs. 6, 7). A maximum likelihood tree based on nonsynonymous SNPs in these *H. spontaneum* genotypes plus those in the elite cultivated barley genotypes revealed three discreet clades each containing one elite line haplotype (Supplementary Figure 7). We then genotyped the L189P polymorphism in 184 georeferenced landraces and found that 7 genotypes (<4%), mostly located in western Europe, contained the P189 substitution (Supplementary Figs. 6, 7). Strikingly, this frequency increased to 35% in the 131 elite genotypes used for GWAS (Supplementary Fig. 7). This increase, which occurs across all branches of the cultivated genepool, is symptomatic of what would be expected for a locus that is currently in the process of undergoing directional selection.

## Discussion

Sodium accumulation is a complex trait that can have serious implications for plant performance and survival. By combining high density SNP array and ionomic data collected from the grain of plants grown under nonsaline conditions, we identified haplotypes of *HvHKT1;5* as the major genetic factor determining Na$^+$ content in contemporary two-row spring barley. *HKT1;5* has previously been implicated in conferring a degree of salinity tolerance in wheat and barley through a Na$^+$ exclusion mechanism[23,24]. In high Na$^+$ accumulating genotypes we found that a single SNP generating an L189P amino-acid residue substitution led to severely compromised HvHKT1;5 function. We propose this is likely due to a combination of protein structural changes leading to misfolding, aberrant subcellular localisation, and subsequent degradation, and is compounded by higher transcript abundance of the functional alleles[21]. Intriguingly, a recent study[39] in bread wheat identified the exact same nucleotide substitution found in the high Na$^+$ accumulating HvHKT1;5$_{HAP3}$ in the landrace Mocho de Espiga Branca, TaHKT1;5 (L190P). Like *HvHK1;5* when this allele is expressed in oocytes, elicited currents were similar to water-injected controls with the conductance unaltered when external Na$^+$ concentration was increased. Similarly, this TaHKT1;5 (L190P) was also localised to internal cell structures. In bread wheat the TaHKT1;5 (L190P) was associated with lower Na$^+$ retrieval from the xylem and high shoot Na$^+$ accumulation.

After growing representatives of each haplotype under saline conditions, our observations align with previous findings in barley[19,20], and in rice[25] in response to short-term NaCl stress. However, in the latter, longer term stress (21 days of 40 mM or 80 mM NaCl) led to a 72% decrease in biomass in an *OsHKT1;5* expression mutant compared to wild-type. Consistent with these findings, when Munns et al.[33] backcrossed a functional *Nax2* locus (*TmHKT1;5 A*) from *T. monococcum* into commercial Durum wheat they observed a grain yield increase of 25% compared to the control when grown on saline soils. Both studies parallel the relationship observed between Arabidopsis *AtHKT1* allele and seed number from plants grown under saline conditions; wild-type Arabidopsis Col-0 produced seeds when exposed to moderate salt stress while an *athkt1* knockout mutant was virtually sterile[34]. Together they suggest that HKT1's are critically important for maintaining fitness under saline conditions. Our

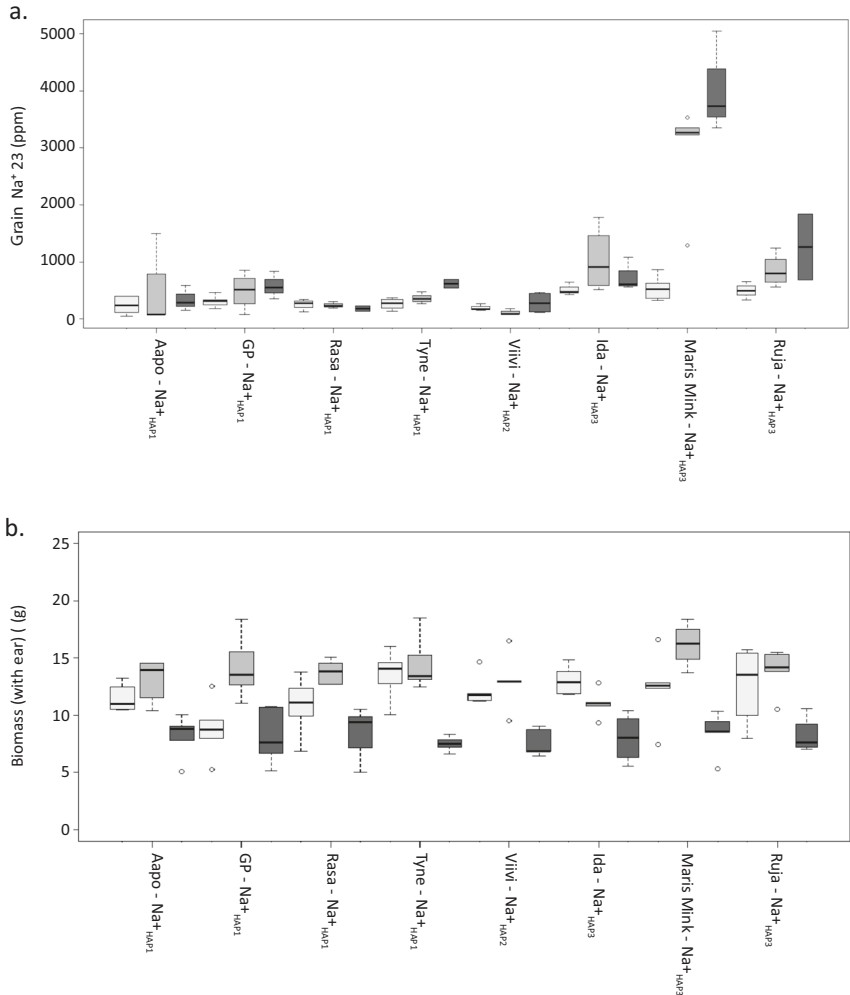

**Fig. 4 Influence of L189P polymorphism in HvHKT1;5 on grain Na+ accumulation and other morphological traits. a** Mature grain Na+ content of barley genotypes exposed to 0, 150, and 250 mM NaCl at the fourth leaf stage of development. **b**. Above ground biomass (including the ear of grain) after barley genotypes were exposed to different concentrations of NaCl at the fourth leaf stage of development. For both **a** and **b** HvHKT1;5 haplotype is shown alongside name of genotype, low grain Na+$_{HAP1}$, Na+$_{HAP2}$, and high grain Na+$_{HAP3}$. White bars indicate 0 mM of NaCl, light grey bars indicate 150 mM NaCl, and dark grey indicates 250 mM NaCl added to plants. For the boxplots the horizontal bar of the boxplot shows the median, the box delineates the first and third quartile, and the whiskers show ±1.5 × IQR, $n = 5$ biologically independent samples.

data, supported by the recent evidence that shoot Na+ accumulation in certain bread wheat genotypes is not negatively associated with plant salinity tolerance[39,40], question this conclusion for barley. Here we show that biomass yield is maintained in lines such as Maris Mink (HvHKT1;5$_{HAP3}$) that have levels of grain (and leaf) Na+ content that would be expected to have a significant negative impact. We conclude that alternative or additional mechanisms must be involved in Na+ tolerance in barley and that Na+ exclusion by HvHKT1;5 per se may be a relatively minor player. This discrepancy could potentially arise because—unlike barley—Arabidopsis, rice, and Durum wheat are all particularly sensitive to saline conditions, pointing to fundamental differences in the roles of HKT1;5 between salt-tolerant and salt-sensitive plants. Recently, HvHKT1;5 RNAi knockdown lines generated in cv.GP (Na+$_{HAP1\_L189}$), were shown to exhibit an increase in shoot biomass compared to wild-type at increasing NaCl concentrations[41]. Somewhat controversially, the authors hypothesise that HvHKT1;5 translocates Na+ from the root to the shoot, and the increase in biomass in the RNAi lines is due to a reduction in translocation of Na+ due to decreased expression of HvHKT1;5. However, the data presented here, in another recent study of HvHKT1;5[21], and in reports from several different crop

species[10–12], indicate that the presence of functionally compromised alleles of HvHKT1;5 leads to an increase in shoot and grain sodium content due to a reduced ability to exclude Na+ from the plant.

The observed increase in the frequency of high Na+ HvHKT1;5$_{HAP3}$ in breeding germplasm from NW Europe returns us to the possible role of Na+ as a functional micronutrient in agriculture and its beneficial effects on plant growth and development, particularly in low K+ environments. While the NW European growing environment is largely devoid of saline soils, K+ deficiency is widespread both in soils with a low clay content and where the annual removal of K+ by high yielding varieties is greater than the amount of K+ applied. In such situations the higher Na+ content of HvHKT1;5$_{HAP3}$ could provide a physiological advantage, for example through use of Na+ as a substitute for K+ in a range of metabolic functions or as a free and abundant osmolyte to reduce leaf water potential, increase (or maintain) transpiration and photosynthesis, and ultimately impact yield. Assuming the latter hypotheses are correct, we cautiously speculate that the increase in frequency of HvHKT1;5$_{HAP3}$ may reflect ongoing positive selection during breeding due to it providing a selective advantage. However, given it is not yet near

fixation in elite genotypes, distinguishing this hypothesis from alternatives such as indirect selection due to linkage drag with another positive trait (i.e. hitchhiking) clearly remains to be tested.

Overall, we conclude that natural allelic variation at *HvHKT1;5* has a strong influence over grain and shoot $Na^+$ homeostasis in barley in both nonsaline and saline environments. A single SNP causing an L189P amino-acid residue substitution likely results in a change in protein structure, that leads to aberrant subcellular localisation, loss of capacity to transport $Na^+$ and consequent reduction in capacity to remove $Na^+$ from the transpiration stream leading to elevated concentrations in the shoots and grain. When grown under salt stress conditions we observed no negative consequences of the exceptionally high concentrations of $Na^+$ found in genotypes containing *HvHKT1;5*$_{HAP3}$ on a range of life history traits, most notably biomass. This questions the widely accepted role of *HvHKT1;5* driven $Na^+$ exclusion in the root as the dominant player in salinity tolerance in barley. Given the identical amino-acid residue substitution (L190P) in *TaHKT1;5D* was recently identified in the bread wheat genotype, Mocho de Espiga Branca, that similarly exhibits atypically high shoot $Na^+$ content and significantly reduced $Na^+$ conductance in TEVC experiments[39], we provide a remarkable example of parallel evolution in two economically important species with potential value in plant breeding.

## Methods

**Phenotypic charecterisation of grain sodium for GWAS.** A collection of 131 contemporary European two-rowed spring barley genotypes, for the purposes of this study our GWAS panel, were grown in a polytunnel in Dundee, Scotland, using standard barley soil and growth conditions. These are as follows: peat = 1.2 m$^3$, sand = 100 l, osmocote exact start = 1.5 kg, osmocote Exact Mini/ 3–4 month/10–12 month =3.5 kg, 2.5 kg each of Lime, Ca, and Mg, celcote = 0.5 kg, perlite = 100 l, and intercept/exemptor = 280 g/390 kg.

We screened the grain sodium concentration of the GWAS panel using Inductively Coupled Plasma Mass Spectrometry (ICP-MS). Barley grains were transferred into Pyrex test tubes (single grain per tube) and weighted. Samples were predigested overnight at room temperature with 1 mL trace metal grade nitric acid Primar Plus (Fisher Chemicals) spiked with indium internal standard followed by digestion in dry block heaters (DigiPREP MS, SCP Science; QMX Laboratories, Essex, UK) at 115 °C for 4 h. Then, 1 mL of hydrogen peroxide (Primar, for trace metal analysis, Fisher Chemicals) was added and samples were digested in dry block heater at 115 °C for 2 h. After cooling down, the digests were diluted to 10 mL with 18.2 MΩcm Milli-Q Direct water (Merck Millipore) and elemental analysis was performed using PerkinElmer NexION 2000 ICP-MS equipped with Elemental Scientific Inc. autosampler, in the collision mode (He). Twenty-one elements (Li, B, Na, Mg, P, S, K, Ca, Cr, Mn, Fe, Co, Ni, Cu, Zn, As, Rb, Sr, Mo, Cd, and Pb) were monitored. The isotopes 23 and 39 were measured for Na and K, respectively. Liquid reference material composed of the pooled digested samples was prepared before the beginning of the sample run and was used throughout the whole samples run. It was run after every ninth sample in all ICP-MS sample sets to correct for variation between and within ICP-MS analysis runs. The calibration standards (with indium internal standard and blanks) were prepared from single element standards solutions (Inorganic Ventures; Essex Scientific Laboratory Supplies Ltd, Essex, UK). Sample concentrations were calculated using external calibration methods within the instrument software. Further data processing was performed in Microsoft Excel. For each genotype Na was measured in five single grains per sample digested, equating to five biological replicates. From these data BLUPs were predicted using GenStat (15th edition).

**DNA extraction, 50k iSelect genotyping, and GWAS.** DNA from 7-day-old leaves was extracted for all genotypes using the QIAamp kit (Qiagen) on the QIAcube HT (Qiagen) using default settings. All samples were genotyped using the 50k iSelect SNP array as described in[18]. GWAS was carried out on adjusted variety means using the EMMA algorithm, a kinship matrix derived using Van raden, and setting the PCA to true in GAPIT[42] with R version 3.5.2[43]. False discovery rates as calculated in GAPIT using Benjamini-Hochberg (1995) FDR-controlling procedure were used to account for the large number of SNPs used in this analysis. We anchored regions of the genome, which were significantly associated with $Na^+$ content to the physical map of the barley sequence to provide annotations for genes within these regions[22,28]. Linkage disequilibrium (LD) was calculated for regions of the genome containing significant associations between pairs of markers using a sliding window of 500 markers and a threshold of R2 < 0.2 using Tassel v5[44] to allow us to identify local blocks of LD, facilitating a more precise delimitation of

QTL regions. We anchored regions of the genome containing markers that passed the FDR to the physical map and then expanded this region using local LD derived from genome-wide LD analysis as described above.

**Resequencing *HvHKT1;5* and sequence alignment.** For the 131 genotypes of the GWAS panel we PCR amplified and Sanger sequenced the coding sequence of *HvHKT1;5*. We used the primers listed in Supplementary Data 4 to resequence this gene. DNA was amplified and cleaned up prior to Sanger sequencing on an ABI3100 capillary sequencer using reaction mixes and conditions described[45]. Sequences were aligned in Geneious version 9.0.2(Biomatters Ltd). Haplotype networks were produced using PopART version 1.7[46]. Orthologs of HvHKT1;5 were identified using the blastx function at NCBI, and the sequences retrieved aligned in Geneious version 9.0.2 (Biomatters Ltd) using MUSCLE with default settings.

**RNA extraction and cDNA synthesis.** Materials detailed in Fig. 2a were sampled, snap frozen and stored at −80 °C for RNA extraction. The root tissue was ground to fine powder on 2010 Geno/Grinder® (SPEX SamplePrep) at 1200 RPM for 30 seconds, and RNA was extracted from the tissue powder by using Direct-Zol RNA MiniPrep (Zymo Research) according to the manufacturer's protocol. Final elution was performed with 40 μL DNA/RNAase-Free water supplied with the kit and the eluted RNA was subsequently quantified using ND-1000 Spectrophotometer (NanoDrop Technologies). cDNA synthesis was then performed on 500 ng RNA by using High Capacity cDNA Reverse Transcription Kit (Thermo Fisher Scientific) according to the manufacturer's instruction in a 20 μL reaction and stored at −20 °C until use.

**RNA extraction and qPCR of *HvHKT1;5*.** Root and shoot tissue from 12-day-old roots and shoots were collected in sections as described[47] from genotypes grown in the same polytunnel as described above, in the same conditions for RNA extraction. Each of the four Biological reps consisted of tissue collected from 15 individual plants. cDNA was synthesised using RNA to cDNA EcoDry™ Premix (Double Primed) (Takara) using standard conditions and used for qPCR. qPCR and the analysis of the subsequent data was carried out as described[47] using 3 housekeeping genes, α–tubulin, GAPDH, and HSP70. Primer sequences and annealing temperatures are provided in Supplementary Data 4.

**Characterisation of diversity of HKT1;5.** Species orthologs of HKT1;5 were identified using the blastx function at NCBI, and the protein sequences retrieved were aligned in Geneious version 9.0.2 (Biomatters Ltd) using MUSCLE with default settings. For the 73 *H. spontaneum* genotypes, DNA was extracted, *HvHKT1;5* amplified and Sanger sequenced as described above. For the landraces, the L189P SNP was genotyped in 184 georeferenced genotypes[38] using primer pair 3 in Supplementary Data 4 and PCR-sequencing conditions described above. The geolocation data for these genotypes is available[37].

**In situ PCR.** Barley roots in situ PCR was followed by Athman et al.[48] with the following modifications. Root cross sections (from maturation zone) were 60 μm obtained using Vibrating Microtome 7000 Model 7000smz-2 (Campden Instruments Ltd.). Thermocycling conditions for the PCR were: initial denaturation at 98 °C for 30 seconds, 35 cycles of 98 °C for 10 seconds, 59 °C (for *HvHKT1;5*), or 57 °C (for *Hv18S*) for 30 seconds, 72 °C for 10 seconds, and a final extension at 72 °C for 10 min. Gene specific primers for *HvHKT1;5* and *Hv18S* (positive control) are shown in Supplementary Data 4.

**Characterisation of HvHKT1;5 in oocytes.** Methods for functional characterisation of HvHKT1;5 variants in *Xenopus laevis* oocytes were as described previously[30,32]. Haplotype and engineered variants of HvHKT1;5 were synthesised by GenScript (Piscataway, NJ, USA) and fragments were inserted into a gateway enabled pGEMHE vector. Nucleotides encoding HvHKT1;5 N57S, P189L, V416I and S438N were modified by site-directed mutagenesis PCR using Phusion®High-Fidelity DNA Polymerase (New England Biolabs, Massachusetts, USA). pGEMHE constructs were linearized using sbfI (New England Biolabs, Massachusetts, USA) followed by ethanol precipitation. Complimentary RNA (cRNA) was transcribed using using the Ambion mMESSAGE mMACHINE kit (Life Technologies, Carlsbad, CA, USA), 23 ng of cRNA (in 46 nL) or equal volumes of RNA-free water were injected into oocytes, followed by an incubation in ND96 for 24–48 h before recording. Membrane currents were recorded in the HMg solution (6 mM $MgCl_2$, 1.8 mM $CaCl_2$, 10 mM MES and pH 6.5 adjusted with a TRIS base) ±$Na^+$ glutamate and/or $K^+$ glutamate as indicated. All solution osmolarities were adjusted using mannitol at 220–240 mOsmol kg$^{-1}$[31,49].

**Transient expression of HvHKT1;5 in Nicotiana benthamiana.** Transient expression of fluorescent fusion proteins was performed as described in detail[26]. In brief, *HKT1;5* coding sequences were recombined into *pMDC43* to generate N-terminally GFP-tagged proteins. For colocalisation studies, nCBL1-RFP was used as a PM-marker[49]. All constructs were transformed into *Agrobacterium tumefaciens* strain Agl-1. Agroinfiltration was performed on fully expanded leaves of 4- to

6-week-old *Nicotiana benthamiana* plants. After 2 days, leaf sections were imaged using a Nikon A1R Confocal Laser-Scanning Microscope equipped with a 633-water objective lens and NIS-Elements C software (Nikon Corporation). Excitation/emission conditions were GFP (488 nm/500–550 nm) and RFP (561 nm/ 570–620 nm).

**Construction of 3D molecular models of HvHKT1;5$_{HAP3\_L189}$ and HvHKT1;5$_{HAP3\_P189}$ in complex with Na$^+$.** The most suitable template for cereal HKT1;5 transporter proteins was the *B. subtilis* KtrB K$^+$ transporter (Protein Data Bank genotype 4J7C, chain I) as previously identified. In KtrB, K$^+$ was substituted by Na$^+$ during modelling of all HKT1;5 proteins. 3D models of HvHKT1;5$_{HAP3\_L189}$ and HvHKT1;5$_{HAP3\_P189}$ in complex with Na$^+$ were generated in Modeller 9v19[50] as described previously[51] incorporating Na$^+$ ionic radii[52] taken from the CHARMM force field[53], on the Linux station running the Ubuntu 12.04 operating system. Best scoring models (from an ensemble of 50) were selected based on the combination of Modeller Objective Function[54] and Discrete Optimised Protein Energy term[55] PROCHECK[56], ProSa 2003[57] and FoldX[58]. Structural images were generated in the PyMOL Molecular Graphics System V1.8.2.0 (Schrödinger LLC, Portland, OR, USA). Calculations of angles between selected α-helices in HvHKT1;5 models were executed in Chimera[59] and evaluations of differences ($\Delta\Delta G = \Delta Gmut - \Delta Gwt$) in Gibbs free energies was performed with FoldX[58]. Sequence conservation patterns were analysed with ConSurf[60,61] based on 3D models of HvHKT1;5 transporters.

Evaluations of stereo-chemical parameters indicated that the template and HvHKT1;5 models had satisfactory parameters as indicated by Ramachandran plots with two residues positioned in disallowed regions, corresponding to 0.5% of all residues, except of G and P. Average G-factors (measures of correctness of dihedral angles and main-chain covalent bonds) of the template, and HvHKT1;5$_{HAP3\_L189}$ and HvHKT1;5$_{HAP3\_P189}$ models, calculated by PROCHECK (0.06, −0.07, and −0.07, respectively), and ProSa 2003 z-scores (measures of C$^\beta$-C$^\beta$ pair interactions of −9.0, −5.6, and −6, respectively), indicated that template and modelled structures had favourable conformational energies.

**Plant material and growth conditions**. Eight barley genotypes (*Hordeum vulgare*) with variation in the key functional SNP (P189L) in *HvHKT1;5* were grown for screening leaf and grain Na$^+$ accumulation under salt stress conditions. This set of genotypes consist of at least three representative barley genotypes of each allele of P189L in HvHKT1;5 characterised in the elite germ in plasm within this study, cv. Golden Promise (GP) (Na$^+_{HAP1\_L189}$), Aapo (Na$^+_{HAP1\_L189}$), Rasa (Na$^+_{HAP1\_L189}$), and Tyne (Na$^+_{HAP1\_L189}$), Viivi (Na$^+_{HAP2\_L189}$), Ida (Na$^+_{HAP3\_P189}$), Maris Mink (Na$^+_{HAP3\_P189}$), Ruja (Na$^+_{HAP3\_P189}$). Three germinated seeds from each genotype were sown in a $10 \times 10$ cm pot filled with a standard cereal compost mix as described above. Eight replicates were sown per genotype in a randomized design. Every 24 pots were randomized in a plastic gravel tray ($56 \times 40 \times 4$ cm), and in total 12 trays were placed in the glasshouse under long-day conditions (light:dark, 16 h:8 h, 18 °C:14 °C). At sowing, the soil moisture and weight of all pots ranged between 31–35% (w/w) and between ~380–385 g. The three seedlings in each pot were thinned to one at the emergence of 2nd leaf.

Before applying the salt treatment soil moisture content in all pots was controlled to around 25% (w/w) to provide larger NaCl uptake capacity. At the emergence of the 4th leaf, a single salt treatment (150 or 250 mM NaCl) was applied directly into each tray in a 2 L volume. The same volume of water was added to the control trays. The fully expanded 5th leaf was harvested for Na$^+$ content analysis using ICP-MS as described above to evaluate the strength of salt treatment (Supplementary Data 5). Plants were harvested at maturity and the Na$^+$ contents of grains and 5th leaves of five of the eight replicates quantified by ICP-MS. Detailed description of the analysis of the resulting data is included in Supplementary Data 1.

**Phylogenetic analysis of HvHKT1;5**. Phylogenetic analysis was conducted using MEGA 7.0.14[62]. For all four datasets we first did a Model selection using maximum likelihood. Sites containing gaps in more than 5% of the data were removed from the analysis. For the elite cultivated barley and the landraces, the GTR + G + I model was selected (General Time Reversible model, Gamma distributed with Invariant Sites). For the *H. spontaneum* genotypes a HKY + G model was selected (Hasegawa–Kishino–Yano model, Gamma distributed). For the *H. sponatneum* tree with only nonsynonymous SNPs (Figure 7A) the Jukes–Cantor model was selected. The phylogenetic tree was done using Maximum Likelihood method with 100 bootstrap replications and the above described models accordingly. Bootstrap values above 60% (represented as 0.6) are shown on the tree.

**Statistics and reproducibility**. Analysis of the phenotypic data collected after plants were exposed to 0, 150, or 250 mM was carried out using ASReml-R and Genstat. Significance was tested at the 5% level, applying either Tukeys or Bonferroni least significant difference to test for differences between treatment levels. A detailed description of the analysis of this dataset is provided in the Supplementary dataset section of the Supplementary Information.

**Reporting summary**. Further information on research design is available in the Nature Research Reporting Summary linked to this article.

## Data availability

NCBI accession numbers for Sanger sequence data corresponding to *HvHKT1;5* which have no restrictions are available in Supplementary Data 1 and 3. Genotyping data used for GWAS and all other relevant data not provided in this manuscript already are available from the corresponding authors upon request.

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

## Acknowledgements

R.W., M.S., and M.M. acknowledge support from ERC project 669182 'SHUFFLE' to R.W. K.H., P.S., M.B., J.R., and R.W. acknowledge the Rural & Environment Science & Analytical Services Division of the Scottish Government. C.B., J.Q., and Y.Q. acknowledge support from Rutherford Fund Strategic Partner Grants 2018 - Award Reference: RF-2018-30 to C.H. and R.W. S.R. is grateful for financial assistance from The Australian Research Council Industrial Transformation Research Hub for Wheat in a Hot and Dry Climate (IH130200027), the Grains Research and Development Corporation (ACP00009) and The Waite Research Institute, University of Adelaide. C.B. acknowledges support from the Grains Research and Development Corporation (GRDC) and the Australian Research Council (FT180100476). We are thankful to the Australian Research Council for funding through CE140100008 to M.G. (J.Q. and Y.Q.), FT180100476 to C.S.B. and DE160100804 to S.W. (A.S.). D.E.S. acknowledges support from BBSRC Grant BB/L000113/1. MH acknowledges financial support from the Huaiyin Normal University, China.

## Author contributions

R.W., D.E.S., M.G., C.B., K.H., J.Q., J.R., and S.R. designed experiments. K.H., J.Q., S.W., Y.Q., P.S., A.S., M.M., and P.F. carried out experiments. K.H., M.H., M.S., H.O., and M.B. analysed data. The manuscript was written by K.H. and R.W. with contributions from all other authors.

## Competing interests

The authors declare no competing interests.
