## [Peer Review file · Communications Biology]

Reviewers' comments:

Reviewer #1 (Remarks to the Author):

Dear Authors,

The paper is well-written and described the hypothesis with many evidences that HKT1;5 is has a positive impact on the traits under non-saline conditions. The gene was detected by natural variation using GWAS then validated by many molecular approaches.

it was mapped before but this is the first time to provide confirmation that this gene is very interesting for breeding, geneticists and agronomist.

Even though, there are many points that would be nice to be considered in the revised version:

The abstract was only used to describe the results and the conclusion was clear. e.g. used "we" three times in the abstract found it strange.

It was not clear what does L189P and P189 mean?

there are many general statements in the introduction, e.g, provides a range of benefits, which type of benefits and please cite the references.

in such cases....

the panel was not well-described in the text, e.g. the population structure, LD....and why the used this population with only two-rowed barley? what do you think about the six-rowed? is there a difference between them in terms of Na⁺ content? are you sure the collection is two-rowed or could be intermedium barley?

why did you use morex (six-rowed) barley to represents HAP3? what does it mean?

the conclusion at the end of results is not well connected with the GWAS, i.e can you explain more about the allele(s) and marker (SNP) from 50K or 9K.

the discussion part needs to be carefully written, e.g. "While the NW European growing environment is largely devoid of saline soils it is subject to periodic spring/summer droughts and depleting levels of soil K⁺ due to annual offtake from high yielding varieties being greater than the amount of K⁺ applied. "

why did you claim about the drought in NW Europe? is it correct? I agree with you that the allelic variation at HvHKT1;5 is very important BUT not for the European area. It would be much more suitable for the area with drought e.g. middle east.

another point, how it could be that Na⁺ can replace the role of K⁺. did you test it?

Regarding to the statement "However, given it is not yet near fixation in elite genotypes, distinguishing this hypothesis from

alternatives such as selection due to linkage disequilibrium with another positive trait clearly remains to be tested." Since you used UK genotypes, is it possible to test it on larger elite material from EU or worldwide?

I found the description for the growing condition needs more information, e.g. fertilizers and the amount of each component.

Did you test the Na⁺ at different grain stage to check which stage is critical for Na⁺ accumulation and expression of the gene?

Reviewer #2 (Remarks to the Author):

This paper describes a nice GWAS study in barley on sodium content, where they identify the HKT1;5 gene as a gene that is associated with sodium content. The investigators show that haplotype 3 of this gene is associated with high sodium, and in a series of experiments in ion conductance in oocytes they identify a specific amino acid substitution that appears causal to compromised sodium transport. They also show increased frequency of this hap 3 in elite germplasm, which they suggest may be due to selection for increased sodium content in potassium-poor soils.

This is a comprehensive GWAS study with some very interesting results. In general, it is fine but there are a few issues:

1. It would be nice in the results section to have more details of the GWAS mapping – how much population structure was there in the panel, did they define a threshold for significance (it doesn't say in the results and the figure shows no threshold line).
2. The selection argument – this is interesting but should be done cautiously. One cannot rule out that selection is acting on a linked locus and that the hap 3 allele is just hitchhiking. This should be pointed out as a possibility.

Reviewer #3 (Remarks to the Author):

Review of "A Grain of Salt"

The authors present interesting GWAS data that may support the contention that sodium could be a positive feature for cereal biomass, not just a detriment as is commonly studied.

My first comment is that the authors move directly into the GWAS results without discussing the applicability of GWAS to the current system. Is population structuring in barley sufficiently low, or if not, factored out somehow in the current experimental design, for GWAS to be effective? I am not familiar with this particular organismal system, but I know that population structuring, such as exists in humans for reasons of genetic drift alone, affects the applicability of GWAS. It is possible, in other words, that the associations found have to do with population structure alone as opposed to biological associations such as the influence of natural selection. While the gene highlighted does appear to be a very predictable candidate, background on barley population structure as a possible confounding factor should be described.

Methodologically, the paper seems largely sound, and it presents an array of exciting results. As a bioinformatics-focused person, I noticed the following potentially serious issue that I think could far better analyzed: "A neighbour joining tree based on nonsynonymous SNPs in these H. spontaneum genotypes plus those in the elite cultivated barley genotypes revealed three discreet clades each containing one elite line haplotype." That tree is presented in Suppler. Fig. 7. First of all, neighbor joining is less well suited to SNP-based phylogenetic inference than an approach that includes optimization using a model of molecular evolutionary processes. I would recommend a maximum likelihood approach be used. Second, even with neighbor joining, the investigators need to evaluate the robustness of conclusions (branchings) using a resampling approach such as bootstrapping. Third, I can't find how the SNP variation was transformed to generate the tree in the first place. Neighbor joining is a distance-based method, so some information on how distances were calculated is needed. If the authors switch to likelihood, then they need to describe methods precisely so they are replicable. It seems likely that the tree topology may remain

similar, with clade separation by haplotype, but proper phylogenetic methods (at minimum - bootstrap support calculations) should be employed.

Other comments...

While the title is cute, I suggest that something more descriptive be used instead, perhaps including "barley" and "HKT1;5" ;)

The abstract is dry and could be improved with a statement of the problem at the outset. Poor context as it stands.

Reviewer #1 (Remarks to the Author):

The abstract was only used to describe the results and the conclusion was clear. e.g. used "we" three times in the abstract found it strange. It was not clear what does L189P and P189 mean? *Abstract has been re-written*

there are many general statements in the introduction, e.g, provides a range of benefits, which type of benefits and please cite the references. in such cases.... *The following sentence on page 4 elaborates on this point: 'In such cases, Na⁺ appears capable of substituting for many of the essential roles that K⁺ ions play in plant nutrition, including enzyme activation and particularly, osmoregulation¹.'*

the panel was not well-described in the text, e.g. the population structure, LD....and why the used this population with only two-rowed barley? what do you think about the six-rowed? is there a difference between them in terms of Na⁺ content? are you sure the collection is two-rowed or could be intermedium barley? *Numerous publications – including several from ourselves (see Darrier et al 2019 – referenced in the text - for a recent summary) report that barley germplasm is largely differentiated on the basis of spring vs. winter growth habit and 2- vs. 6-row type. We prioritise using only one 'group' in GWAS as it removes issues associated with population structure. We use 2-row spring barley because it is the major classification in in NW Europe and does not require vernalisation (note - the 2-rows are not intermedium – there are no intermedium elite barley lines). There are fewer 6-row elite spring types as it is not a class of barley widely cultivated in Europe (six-row winters are). All wild ancestral genotypes are 2-rowed and 6-row types appeared soon after the domestication of the species. They remain fully interfertile with the 2-row types. Many of the genotypes included in the survey of European landraces from Russell et al (also referenced in the text) were six-row and show the same non-functional allele as found in the 2-row springs. We summarise this on Page 5*

why did you use morex (six-rowed) barley to represents HAP3? what does it mean? *Choice of Morex and Golden Promise was simply pragmatic based on both having available pseudomolecule scale genome assemblies. One is the genome reference, the other the functional genomics reference. We refer to this now on Page 6*

the conclusion at the end of results is not well connected with the GWAS, i.e can you explain more about the allele(s) and marker (SNP) from 50K or 9K. *The 50K marker and distance from HorVu now provided on page 5*

the discussion part needs to be carefully written e.g. "While the NW European growing environment is largely devoid of saline soils it is subject to periodic spring/summer droughts and depleting levels of soil K⁺ due to annual offtake from high yielding varieties being greater than the amount of K⁺ applied. "why did you claim about the drought in NW Europe? is it correct? I agree with you that the allelic variation at HvHKT1;5 is very important BUT not for the European area. It would be much more suitable for the area with drought e.g. middle east. *This is the crux of our argument. The issue with NW Europe is that the soils are not saline and are generally pretty wet (by comparison to true dryland agricultural areas). Our point is that under intensive agricultural systems these soils can often be depleted of K⁺ (or K⁺ is not bioavailable). We argue that in such cases the increased effective uptake of Na (actually decreased exclusion) by lines that do not exclude Na efficiently can have a positive effect by substituting for some of the roles usually played by K⁺ - such as providing a cheap/free osmoticum that helps maintain transpiration and photosynthesis. Increased ability to pull water from the soil will of course also have a positive impact during periodic droughts (which are*

increasingly common in certain high production areas). The referee is correct - the compromised allele may not have the same impact under long term drought – but this is something we are keen to test. Our hypothesis is supported by the low frequency and distribution of the compromised haplotype in barley landraces and *H. spontaneum* and its increase in frequency in germplasm grown in non-saline environments. Information about the low variation at L189P in these collections is provided on page 10. The large number of haplotypes for HvHKT1;5 observed in *H. spontaneum* is described on page 10.

another point, how it could be that Na⁺ can replace the role of K⁺. did you test it? *We did not test this directly (I don't think anyone has other than in physiological experiments that measure integrated outputs such as biomass). We refer to numerous such publications that allude to the potential of K⁺ being substituted by Na⁺. We would prefer to use true isogenic lines to avoid confounding by genotype and are making such lines now using both CRISPR and targeted mutagenesis – but they will not be ready for use for some time.*

Regarding to the statement "However, given it is not yet near fixation in elite genotypes, distinguishing this hypothesis from alternatives such as selection due to linkage disequilibrium with another positive trait clearly remains to be tested." Since you used UK genotypes, is it possible to test it on larger elite material from EU or worldwide? *Yes of course this is possible and we will likely endeavour to do this as our work progresses. We feel however that 131 cultivars is probably a good reflection of the elite gene pool – the most important set to look at would be the most recently released varieties (see Page 13)*

I found the description for the growing condition needs more information, e.g. fertilizers and the amount of each component. *Modified on Page 14*

Did you test the Na⁺ at different grain stage to check which stage is critical for Na⁺ accumulation and expression of the gene? *No, the methods describe the tissues which we used to quantify HvHKT1;5 expression on page 16.*

Reviewer #2 (Remarks to the Author):

1. It would be nice in the results section to have more details of the GWAS mapping – how much population structure was there in the panel, did they define a threshold for significance (it doesn't say in the results and the figure shows no threshold line) *.The threshold isn't included in the text to keep things concise however it is in the legend for Figure 1 and we have added the FDR adjusted P value on page 5*

2. The selection argument – this is interesting but should be done cautiously. One cannot rule out that selection is acting on a linked locus and that the hap 3 allele is just hitchhiking. This should be pointed out as a possibility. *We added the term 'cautiously' and emphasised alternatives to direct selections - see Page 13*

Reviewer #3 (Remarks to the Author):

My first comment is that the authors move directly into the GWAS results without discussing the applicability of GWAS to the current system. Is population structuring in barley sufficiently low, or if not, factored out somehow in the current experimental design, for GWAS to be effective? I am not familiar with this particular organismal system, but I know that population structuring, such as exists in humans for reasons of genetic drift alone, affects the applicability of GWAS. **It is possible, in other words, that the associations found have to do with population structure alone as opposed to**

biological associations such as the influence of natural selection. While the gene highlighted does appear to be a very predictable candidate, background on barley population structure as a possible confounding factor should be described. *See comments above about population structure and reasons for choosing the spring 2-row gene pool for our study. Page 5 includes references about population structure and why we selected these accessions.*

Methodologically, the paper seems largely sound, and it presents an array of exciting results. As a bioinformatics-focused person, I noticed the following potentially serious issue that I think could far better analyzed: "A neighbour joining tree based on nonsynonymous SNPs in these H. spontaneum genotypes plus those in the elite cultivated barley genotypes revealed three discreet clades each containing one elite line haplotype." That tree is presented in Suppler. Fig. 7. First of all, neighbor joining is less well suited to SNP-based phylogenetic inference than an approach that includes optimization using a model of molecular evolutionary processes. I would recommend a maximum likelihood approach be used. Second, even with neighbor joining, the investigators need to evaluate the robustness of conclusions (branchings) using a resampling approach such as bootstrapping. Third, I can't find how the SNP variation was transformed to generate the tree in the first place. Neighborjoining is a distance-based method, so some information on how distances were calculated is needed. If the authors switch to likelihood, then they need to describe methods precisely so they are replicable. It seems likely that the tree topology may remain similar, with clade separation by haplotype, but proper phylogenetic methods (at minimum - bootstrap support calculations) should be employed. *We have carried out maximum likelihood analysis, added bootstrap support and updated all relevant figures and the methods section.*

While the title is cute, I suggest that something more descriptive be used instead, perhaps including "barley" and "HKT1;5" ;). *We changed the title to the following: 'Natural variants of HvHKT1;5 regulate sodium content in barley.'*

The abstract is dry and could be improved with a statement of the problem at the outset. Poor context as it stands. *As written above – we rewrote the abstract.*

REVIEWERS' COMMENTS:

Reviewer #1 (Remarks to the Author):

I'm satisfied with the responses in the revised version of manuscript.

Authors considered all my suggestions and comments.

Reviewer #2 (Remarks to the Author):

The authors have addressed my concerns.